# Immune and Reproductive Biomarkers in Female Sea Urchins *Paracentrotus lividus* under Heat Stress

**DOI:** 10.3390/biom13081216

**Published:** 2023-08-04

**Authors:** Alessandra Gallo, Carola Murano, Rosaria Notariale, Davide Caramiello, Elisabetta Tosti, Stefano Cecchini Gualandi, Raffaele Boni

**Affiliations:** 1Department of Biology and Evolution of Marine Organisms, Stazione Zoologica Anton Dohrn, Villa Comunale, 80121 Naples, Italy; alessandra.gallo@szn.it (A.G.); rosaria.notariale@szn.it (R.N.); elisabetta.tosti@szn.it (E.T.); 2Department of Integrative Marine Ecology, Stazione Zoologica Anton Dohrn, Villa Comunale, 80121 Naples, Italy; carola.murano@szn.it; 3Unit Marine Resources for Research, Stazione Zoologica Anton Dohrn, Villa Comunale, 80121 Naples, Italy; davide.caramiello@szn.it; 4Department of Sciences, University of Basilicata, Via dell’Ateneo lucano, 10, 85100 Potenza, Italy

**Keywords:** coelomic fluid, coelomocytes, eggs, FRAP, TTL, myeloperoxidase, protease activity, intracellular pH, reactive nitrogen species

## Abstract

The functioning of the immune and reproductive systems is crucial for the fitness and survival of species and is strongly influenced by the environment. To evaluate the effects of short-term heat stress (HS) on these systems, confirming and deepening previous studies, female sea urchin *Paracentrotus lividus* were exposed for 7 days to 17 °C, 23 and 28 °C. Several biomarkers were detected such as the ferric reducing power (FRAP), ABTS-based total antioxidant capacity (TAC-ABTS), nitric oxide metabolites (NO_x_), total thiol levels (TTL), myeloperoxidase (MPO) and protease (PA) activities in the coelomic fluid (CF) and mitochondrial membrane potential (MMP), H_2_O_2_ content and intracellular pH (pH_i_) in eggs and coelomocytes, in which TAC-ABTS and reactive nitrogen species (RNS) were also analyzed. In the sea urchins exposed to HS, CF analysis showed a decrease in FRAP levels and an increase in TAC-ABTS, TTL, MPO and PA levels; in coelomocytes, RNS, MMP and H_2_O_2_ content increased, whereas pH_i_ decreased; in eggs, increases in MMP, H_2_O_2_ content and pH_i_ were found. In conclusion, short-term HS leads to changes in five out of the six CF biomarkers analyzed and functional alterations in the cells involved in either reproductive or immune activities.

## 1. Introduction

In this century, climate change is primarily responsible for biodiversity loss [1], significantly affecting the functions of terrestrial and aquatic organisms [2,3,4]. In particular, global warming, which is mainly due to the increase in anthropogenic emissions of greenhouse gases and the burning of fossil fuels [5,6], represents the gravest threat to marine life since the average seawater temperature is increasing rapidly, reaching new records [7], and is predicted to continue increasing [8]. Indeed, seawater warming has been estimated to cause harmful effects greater than those induced in terrestrial environments [9]. Water temperature is one of the most important abiotic factors controlling the distribution, morphology and behavior, as well as affecting the physiological performance, of marine organisms. Each species has its own optimum temperature range within which its physiological performance is optimal [10]; thereby, any deviation from this range may reduce its fitness [11,12].

The sea urchin *Paracentrotus lividus* (Lamarck, 1816) is an ecologically important species, which plays key structuring and functioning roles in marine ecosystems [13] and is constantly exposed to environmental pressure, including changes in temperature, to which it is very sensitive. Since the optimal temperature for the reproduction and development of Mediterranean *P. lividus* is 17–20 °C, the warming of seawater temperature may be a potential threat to reproductive processes. Negative impacts on the reproduction of marine organism are widely reported when the environmental temperatures increase beyond their thermal optimum [14]. In particular, a decrease in reproductive success in response to seawater temperature increase has been linked to female and male gamete quality impairment, such as egg morphology alteration and sperm concentration, and motility reduction in teleost fishes and mussels [15,16].

In order to survive in the context of global warming resulting in increased seawater temperature, marine organisms have to maintain functional homeostasis and redox status of their physiological functions [17]. Sea urchins have a sophisticated immune system, whose primary effectors are the coelomocytes, i.e., a heterogeneous population of freely moving cells mainly contained in the coelomic fluid (CF), including red and white amoebocytes, vibratile cells and phagocytes. These cells mediate immune responses and maintain physiological homeostasis [18,19]. The coelomocyte number and their cellular activity represent a good indicator of the immunological status of an organism and its ability to defend itself from pathogens [20]. Moreover, the immune cells of *P. lividus* have been proven to be a good bioindicator of abiotic stress such as seawater temperature increase [21]; thereby, they are a useful tool for marine global change research [22].

The sea urchin *P. lividus* can be found in marine areas exhibiting temperatures ranging from 4 to 30 °C [23], with fluctuations of up to 15 °C [24]. The thermal needs involved in the growth, activity, and gonadal development of *P. lividus* are differentiated and linked to seasonal variations that play a significant impact on the reproduction and fitness of this species [23]. In the framework of the actual climate change scenarios, the present study aimed to assess the impact of temperature gradients by measuring several biomarkers in coelomic fluid related to redox status, inflammation and innate immunity. In addition, some physiological cell parameters related to mitochondrial activity, intracellular reactive oxygen species (ROS), nitric oxide (NO) metabolites, total antioxidant capacity and intracellular pH have been comparatively evaluated either in the female gametes or in immune cells. These are used as a sensitive warning system to detect the impact of warming seawater in female *P. lividus*. Studies in this species, in fact, demonstrate that the female shows a higher level of immune activity than the male [20]. This investigation on the reproduction and immune system of echinoderms would be extremely useful for understanding the impact of climate change on basic biological processes essential for the persistence of marine populations.

## 2. Materials and Methods

### 2.1. Materials

Unless otherwise indicated, all materials used for this study were purchased from Merck (Merk Life Science, Milan, Italy) and cell culture tested.

### 2.2. Animal Collection and Breeding Care

Adult individuals of *P. lividus* were collected in the Gulf of Naples in the middle of breeding season (April–May 2023) [25] from an area not privately owned nor protected, according to the authorization of Marina Mercantile (DPR 1639/68, 19 September 1980, confirmed on 10 January 2000), and transported inside thermal boxes to the Marine Biological Resources Service of Stazione Zoologica Anton Dohrn (SZN). Here, the animals were acclimated for 7 days in tanks (1 animal/5 L) with running natural seawater and under the following conditions: temperature of 17 ± 2 °C, pH 8.2 ± 0.1, salinity 39 ± 0.5 ppm and a photoperiod of 10 h L: 14 h D. During this period, sea urchins were fed with fresh green algae *Ulva* sp. After acclimatization, female sea urchins were screened based on a morphological analysis [26]. The sexual maturity of these animals was assessed by inducing egg spawning (see below) in specimens from the same batch of animals collected. The animals submitted to induced spawning were not subsequently involved in this study.

### 2.3. Experimental Design

A group of 18 animals was randomly divided into three open-circuit seawater tanks assuring up to three complete water changes daily. In the first tank, seawater was maintained at ambient temperature (17 °C, control), corresponding to an optimal reproductive temperature [27], whereas, in the other two tanks, seawater temperature was gradually increased to 23 °C (moderate HS) and 28 °C (severe HS) over 5 days. Then, the experiment started and lasted 7 days. The temperatures and the seawater pH of the three tanks were measured twice a day for the entire duration of the experiment. During the experimental period, the animals underwent food restriction, with only one meal of fresh green algae *Ulva* sp. mid-experiment (day 3), to avoid contamination by feces and food residues that would have involved cleaning operations compromising the rearing environment and causing a further stress source. We are aware that this choice might have generated an additional source of stress to the one analyzed; however, previous experiences in our animal facility and research on this subject [28] demonstrated a remarkable resistance in *P. lividus*, which showed no differences in many metabolic parameters even after one month of starvation. At the end of the experimental period, each sea urchin was measured to test diameter (60.5 ± 17.9 mm without spines) and total weight (69.0 ± 21.3 g); then, coelomic fluid and eggs were collected.

### 2.4. Coelomic Fluid (CF) and Cell Sampling

CF was withdrawn using a 26-gauge needle attached to a 5 mL sterile syringe that was inserted through the peristomium into the coelom cavity [29]. An additional sample of CF was withdrawn with a syringe containing an anticoagulant solution CCM 2X (NaCl 1 M, MgCl_2_ 10 mM, EGTA 2 mM, Hepes 40 mM, pH 7.2), ensuring a 1:1 ratio with the CF withdrawn [30]. Coelomocytes were isolated from CF + CCM via centrifugation at 600× *g* for 10 min at 4 °C.

### 2.5. Egg Collection

Egg spawning was induced by injecting 0.5 M KCl solution into the coelom. After injection, females were inverted over a beaker containing filtered natural seawater (FNSW) and left to release eggs for 15 min. After that, egg morphology was checked under a light microscope.

### 2.6. pH and Osmolarity Evaluation in the Coelomic Fluid (CF)

The CF pH (pH_CF_) and osmolarity were measured with a Jenway 3020 pH meter (Jenway, London, UK) and an osmometer (Digital Osmometer, Roebling, Berlin, Germany), respectively.

### 2.7. Coelomic Fluid (CF) Biochemical Analyses

The ferric reducing power (FRAP) assay, based on the reduction of ferric ions to ferrous ions of the iron-Tris(2-pyridyl)-s-triazine complex, was carried out as described by Benzie and Strain [31]. The assay was calibrated with iron (II) sulphate heptahydrate (FeSO_4_•7H_2_O), and the results are expressed in terms of FeSO_4_•7H_2_O equivalents (µM).

Total antioxidant capacity (TAC), based on the reduction of 2,2′-azinobis-(3-ethylbenzothiazoline-6-sulfonic acid) radical cation (ABTS^•+^), was measured according to Erel [32]. The assay was calibrated with ascorbic acid (AA), and the results are presented as AA equivalents (µM).

Total thiol levels (TTL), based on the interaction of sulfhydryl groups (-SH) with 5,5-dithiobis-(2-nitrobenzoic acid) (DTNB), were measured as described by Hu [33]; the obtained data are presented in terms of TTL concentration (µM).

Nitric oxide radicals (NO) were measured by quantifying their stable metabolites (NO_x_), namely, the sum of nitrite (NO_2_^−^) and nitrate (NO_3_^−^), with Griess reagent, as described by Miranda et al. [34]. The assay was calibrated with sodium nitrate (NaNO_3_) and NO_x_ were reported as NaNO_3_ equivalents (µM).

Myeloperoxidase (MPO) activity was measured according to Quade and Roth [35]. The method is based on MPO-H_2_O_2_ oxidation of 3,3′,5,5′-tetramethylbenzidine hydrochloride (TMB) as a sensitive peroxidase substrate. Results are expressed as optical density measured at 450 nm (OD_450 nm_).

Protease activity (PA) was determined using the azocasein hydrolysis assay, as described by Ross et al. [36], and the results are expressed as a percentage of activity in relation to the positive control, the bovine trypsin.

Analytical results were normalized for the total protein content, which was measured using the Bradford method [37] using a commercial Bradford reagent (cod. B6916).

All the samples analyzed were read with a microplate reader (Model 550, BioRad, Segrate, Italy), except for TTL assays, which were read with a spectrophotometer (SmartSpec 3000 UV/Vis, Bio-Rad, Segrate, Italy).

### 2.8. Coelomocyte Count

Total coelomocyte counts were taken using a Neubauer-improved chamber under a light microscope (Apotome.2, Zeiss, Oberkochen, Germany). Values were corrected taking into account the dilution with anticoagulant. The count values were the average number of coelomocytes observed in 5 microscopic fields. A standard number of 4 × 10^5^ coelomocytes of each animal was seeded in a 96-well plate and incubated for 30 min before adding the fluorescent probes used to evaluate mitochondrial membrane potential (MMP), hydrogen peroxide (H_2_O_2_) content and intracellular pH (pH_i_) in the cells of the three groups under comparison. Photomicrographs by confocal laser scanning microscopy (Zeiss LSM 510) for assessing intracellular localization of each fluorescent dye are shown in Appendix A.

### 2.9. Egg Count and Viability

An aliquot of egg suspension was used for egg counting under a stereomicroscope. Then, egg samples (500 eggs/sample) were stained with the cell-permeable dye (effective for dead- and live-cell nucleic acid staining) bis-benzimide (2 µM H33342, Life technologies, Thermo-Fisher Scientific, Milan, Italy) for 30 min at 18 °C. After two washings, the eggs were incubated with another dye effective for dead cell nucleic acid staining, propidium iodide (12 µM PI, Life Technologies), for 15 min at 18 °C. Then, they were read with a microtiter plate reader (Infinite M1000 PRO; Tecan, Switzerland) by setting the excitation wavelength at 370 and 560 nm and evaluating the emission peak between 400–500 and 600–700 nm for H33342 and PI, respectively. Two control egg samples taken from animals kept at 17 °C were prepared: the first (sample A) was dyed as above and used as a reference to represent 100% viability; the second (sample B) was incubated in glutaraldehyde 2% in FNSW for 30 min at room temperature, stained as above, and was attributed 0% viability (sample B). The equation of the straight line obtained from the ratios of the emission fluorescence intensity peak values of the two fluorescent dyes in samples A and B allowed for transforming the ratios between the two fluorescence intensity peaks of the samples into viability rate.

### 2.10. Mitochondrial Membrane Potential (MMP)

JC1 (5,5′,6,6′-Tetrachloro-1,1′,3,3′-tetraethyl-imidacarbocyanine iodide) (Life Technologies) was used to assess MMP, as previously reported [16,21,38]. In brief, 7.7 µM JC1 was added to 200 µL FNSW containing either 500 eggs or 4 × 10^5^ coelomocytes. After 30 min of incubation at 18 °C, samples were washed twice with NFSW and incubated again for 30 min, as above. Then, egg samples were read with a microtiter plate reader (see above). MMP was expressed as the ratio between the absolute fluorescence intensity emission peaks at ~595 (F_0_B) and ~535 (F_0_A) nm wavelengths. A positive control was prepared by treating pre-loaded JC-1 eggs with 2 µM CCCP (carbonyl cyanide 3-chlorophenylhydrazone) [21].

### 2.11. Intracellular Hydrogen Peroxide (H_2_O_2_) Content

The intracellular content of hydrogen peroxide (H_2_O_2_) was assessed by using the 2′,7′-dichlorodihydrofluorescein diacetate (H_2_DCFDA, Life technologies, Thermo-Fisher Scientific, Milan, Italy), as previously described [38,39]. In brief, 5 µM H_2_DCFDA was added to the 200 µL FNSW containing either 500 eggs or 4 × 10^5^ coelomocytes. After 30 min of incubation at 18 °C, samples were washed twice with NFSW and incubated again for 30 min, as above. Then, egg samples were read with a microtiter plate reader (see above). The intracellular H_2_O_2_ levels were expressed in arbitrary units (a.u.) as fluorescence emission peak intensity at ~525 nm wavelength setting the excitation wavelength at 488 nm. Positive controls were prepared by treating control coelomocytes and eggs with 25 μM hydrogen peroxide before staining [21].

### 2.12. Intracellular pH (pH_i_)

Aliquots of eggs (about 500) and coelomocytes (4 × 10^5^) collected from each animal of the three experimental groups were incubated with 5 µM BCECF-AM (2′,7′-Bis-(2-Carboxyethyl)-5-(and-6)-Carboxyfluorescein, Acetoxymethyl Ester) in FNSW for 30 min at 18 °C. Then, they were washed with FNSW and incubated for an additional 30 min at 18 °C to allow de-esterification of the BCECF [38,40]. After BCECF loading, additional samples from the control (17 °C) group were treated with a calibration solution (143 mM KCl, 5 mM HEPES) supplemented with 5 µM nigericin [40] that was, previously, brought to pH values equal to 6.8, 7.2 and 7.6 using HCl and NaOH. These samples were incubated for 30 min. Then, all samples were read with a microtiter plate reader (see above). The ratiometric analyses of the recorded data were performed by dividing the emission intensity at 535 nm after excitation at 490 nm by the emission intensity at 535 nm after excitation at 440 nm. A linear regression analysis of the calibration produced a formula that was used to convert the data to pH_i_ values.

### 2.13. Intracellular Total Antioxidant Capacity (TAC)

Exploiting the decolorization of 2,2′-azinobis-3-ethylbenzothiazoline-6-sulfonic acid (ABTS) radical cation, generated by oxidation of ABTS with hydrogen peroxide in the presence of horseradish peroxidase, by scavenging ability of antioxidants in coelomocytes, it is possible to determine the TAC of coelomocytes [30]. Specifically, the TAC was quantified by measuring the absorbance at 730 nm using as a reference the standard curve of ascorbic acid (1–15 µM). The TAC values were normalized by the total proteins contents that were evaluated using the Bradford assay [37] using a spectrometer (Infinite M1000 PRO) set at 595 nm and bovine serum albumin as standard.

### 2.14. Intracellular Reactive Nitrogen Species (RNS)

Intracellular RNS levels were assessed by employing DAF-DA (4,5-Diaminofluorescein diacetate) (Merck, Milan, Italy), as previously described [41,42]. Briefly, sea urchin coelomocytes (about 1.0 × 10^6^) were incubated for 1 h in the dark at room temperature with 20 µM DAF-DA in 1 mL of CCM anticoagulant and then collected via centrifugation at 8000× *g* for 10 min at 4 °C. Cells were washed three times with CCM1x in order to remove the unreacted probes and stored at −80 °C. Coelomocytes were resuspended in 0.5 mL Tris-HCl buffer 40 mM, pH 7.0, sonicated for 1 min, and centrifuged for 10 min at 8000× *g* at 4 °C. The fluorescence of the supernatant was measured using the microplate reader at ex 495/em 515 nm. Fluorescence values were normalized by subtracting the autofluorescence of unlabeled extracts (coelomocytes incubated with DMSO). Results are expressed as fluorescence intensity (a.u.) and referred to 1 × 10^6^ cells.

### 2.15. Statistical Analysis

Statistical analyses were performed using ANOVA by using Systat 11.0 (Systat Software Inc., San Jose, CA, USA). Data were measured in duplicate or triplicate, whose averages were entered into a spreadsheet. Before the analyses, percentage values were transformed in arcsine. Normal data distributions and homogeneity of variance were assessed by the Shapiro–Wilks test and Levene’s test, respectively. Pair-wise comparisons of the means were performed with Fisher’s Least Significant Differences (LSD) test. The threshold of *p* < 0.05 was used as the minimum level of statistical significance. Data are shown as mean ± standard deviation (SD).

## 3. Results

Through the experiment, little variations were found in the seawater temperatures of the three experimental tanks that ranged around the values set at 17 °C (16.50 ± 0.44 °C), 23 °C (22.56 ± 0.97 °C), and 28 °C (27.69 ± 0.96 °C). However, to simplify, the results will refer to the set temperatures.

### 3.1. Coelomic Fluid (CF) Parameters

The pH_CF_ did not vary among animals exposed to different temperatures (Table 1); however, it significantly (*p* < 0.01) differed from the seawater pH (8.19 ± 0.06). The CF osmolarity values did not significantly differ among experimental groups (Table 1) and were similar to those recorded in the seawater (1123 ± 2 mOsm).

### 3.2. Coelomocytes and Eggs Count

The coelomocyte concentration significantly (*p* < 0.05) increased in the animals exposed to 23 °C with respect to either the control or the animals exposed to 28 °C (24.0 ± 7.9 vs. 14.8 ± 7.9 and 14.0 ± 7.9 × 10^6^ mL^−1^) (Table 1). In all groups, only five out of six animals released eggs, of which the mean total number, however, did not differ significantly between treatments (Table 1).

### 3.3. Coelomic Fluid (CF) Biochemical Analysis

Table 2 reports the mean (±SD) values of the six biomarkers analyzed in CF for assessing redox balance, inflammation and protease activity. The total antioxidant activity, measured by the FRAP method, showed a statistically significant (*p* < 0.05) decrease in the group at 28 °C compared to the group at 17 °C. TAC-ABTS values, on the other hand, showed a significant (*p* < 0.05) increase in the group exposed to 23 °C compared to the group at 17 °C. TTL values significantly (*p* < 0.01) increased only at 28 °C differing significantly from those recorded at 17 and 23 °C. NO_x_ values did not show significant differences between groups. MPO activity increased significantly (*p* < 0.05) at 28 °C with respect to the group at 17 °C. The protease activity significantly (*p* < 0.05) increased at 23 °C compared to the group at 17 °C and further increased (*p* < 0.01) at 28 °C compared to those recorded at either 23 or 17 °C.

### 3.4. Eggs’ Viability

A significant (*p* < 0.05) reduction in viability was found in the eggs recovered from the sea urchins exposed to both 23 °C (95.0 ± 3.1%) and 28 °C (95.6 ± 2.9%) compared to the group at 17 °C (99.6 ± 2.0%).

### 3.5. Mitochondrial Membrane Potential (MMP) in Eggs and Coelomocytes

The mitochondrial activity of eggs and coelomocytes, evaluated as mitochondrial membrane potential (MMP), was significantly affected by HS (Figure 1). In fact, in both cell types, a significant (*p* < 0.01) increase was found in the group exposed to 28 °C compared to the group exposed to 17 °C (in eggs, 2.67 ± 1.01 vs. 1.07 ± 0.16; in coelomocytes, 4.80 ± 2.07 vs. 2.06 ± 1.34). As regards the animals exposed to 23 °C, in eggs, the MMP was similar to that of the group at 17 °C and significantly (*p* < 0.01) different from the group exposed to 28 °C (0.99 ± 0.42 vs. 2.67 ± 1.01); in coelomocytes, instead, the MMP of the group at 23 °C did not differ from those of either the control group or the group exposed to 28 °C.

### 3.6. Hydrogen Peroxide (H_2_O_2_) Content in Eggs and Coelomocytes

In the two cell types, the H_2_O_2_ content showed a significant increase only after exposure at the highest temperature (Figure 1). In particular, in the eggs, the group at 17 °C showed similar values to the group exposed to 23 °C; however, both groups significantly (*p* < 0.05) differed from the group exposed to 28 °C (2206 ± 535 and 2253 ± 555 vs. 2906 ± 825 a.u.). In coelomocytes, the H_2_O_2_ content significantly (*p* < 0.05) increased in sea urchins exposed to 28 °C with respect to the group exposed to 17 °C (149.0 ± 118 vs. 76.3 ± 11 a.u.), whereas the group exposed to 23 °C did not show significant differences with the other groups.

### 3.7. Intracellular pH (pH_i_) in Eggs and Coelomocytes

The pH_i_ showed an opposite trend in the two cell types in the group exposed to 23 °C (Figure 1). In fact, in eggs, the pH_i_ significantly (*p* < 0.01) increased in the group exposed to 23 °C compared to the group exposed to 17 °C (7.35 ± 0.20 vs. 7.73 ± 0.36), whereas, in the group exposed to 28 °C, it did not significantly differ from those of the other groups. In coelomocytes, the pH_i_ significantly (*p* < 0.01) decreased in the group exposed to 23 °C (7.24 ± 0.28 vs. 6.72 ± 0.49), whereas, in the group exposed to 28 °C, it did not significantly differ from those of the other groups.

### 3.8. Reactive Nitrogen Species (RNS) and Total Antioxidant Capacity (TAC) in Coelomocytes

In coelomocytes, by increasing the temperatures, RNS levels progressively increased, with significant (*p* < 0.05) differences between the group at 17 °C and that exposed to 23 °C (12,374 ± 3340 vs. 14,140 ± 1571 a.u.) which further increased (*p* < 0.01) in the group exposed to 28 °C (15,017 ± 3050 a.u.) (Figure 2A). On the other hand, the TAC measured in coelomocytes did not show statistically significant differences between the experimental groups (Figure 2B).

## 4. Discussion

This study relates the response to heat stress in the cells of the immune and reproductive systems by simultaneously evaluating several physiological biomarkers. Different behaviors in some of these functions between these cell types stimulate reflections on the mechanisms underlying thermotolerance and response to stressful conditions. Simultaneous assessment in immune and reproductive cells following environmental stress represents a scientific novelty investigated in previous studies exclusively in chicken and related to oviposition [43] or postnatal immune effects [44]. Another novelty of this study is the analysis of CF in echinoderms using a panel of six biomarkers; five of them disclosed the stress status, sometimes even in its moderate intensity.

The short-term HS did not affect the CF osmolarity in agreement with with the poor ability of many echinoderms to regulate ion concentration in their extracellular fluids [45]. Also, in agreement with a previous study [21], the CF pH did not significantly vary in relation HS occurrence and remained significantly lower than the seawater pH; this is likely due a CO_2_ retention (slow diffusion rate) and the accumulation of acidic metabolites in the CF [46]. In addition, the number of eggs recovered did not seem to be influenced by HS. To our knowledge, similar determinations on female gametes of marine animals have not been previously performed. However, previous studies demonstrated that sperm count was not affected by environmental stress such as after exposure to acidified seawater in either the ascidian *Ciona robusta* [47] or *Mytilus galloprovincialis* [48]. Nevertheless, in the latter species, a reduction in sperm concentration was detected from day 7 to day 30 of HS [16]. On the other hand, differently from a previous study reporting that coelomocyte concentration was not affected by HS [21], in this study, a significant increase was found in the coelomocyte concentration of sea urchin exposed to 23 °C. Coelomocyte concentration may depend on the type of stressors. In fact, it normally increases under pathological conditions or threats from pathogens [49], it seems to be not affected by chemical stressors [22,49] and it decreases after a multi-stressor exposure, such as thermal stress and seawater acidification [50].

Regarding the biochemical parameters analyzed in the CF, most of them describe the redox potential of this biological fluid, whereas the protease activity represents a defense mechanism against pathogens. A wide battery of biomarkers cumulatively assessing different compounds related to the redox status allows us to investigate the biological responses that could escape with the use of a single biomarker [51]. To our knowledge, these biomarkers have never been evaluated in the CF of echinoderms; hence, a comparative analysis of our results will be carried out using other marine invertebrates.

The FRAP assay measures the antioxidant capacity of the biological fluids mainly based on the cumulative antioxidant effect of low-molecular-weight antioxidants (i.e., ascorbic acid, α-tocopherol, uric acid, bilirubin and polyphenolic compounds) and provides an index of ability to resist oxidative damage [31]. The progressive reduction in this biomarker in relation to seawater temperature increase suggests that these antioxidant defense systems are impaired by HS. However, variations in FRAP values may rely on the stress duration. In fact, a significant increase in FRAP was found in other marine invertebrates after a very short HS, as detected in the hemolymph of the tropical mussel *Perna viridis* following 24h of HS [52] or in the corals *Pocillopora damicornis* and *Pocillopora meandrina* exposed for 3 h to temperatures higher than that of their thermal comfort (26.5–27 °C) [53]. On the other hand, a recent study conducted on Zhikong scallop (*Chlamys farreri*) demonstrated that in the extracts of mantle and gill [54], FRAP values significantly increased after 6 h and significantly decreased after 30 days of HS.

Another biomarker useful to evaluate the total capacity of the CF to counteract the radicals and estimate its antioxidant capacity was the TAC based on the ABTS method. The increase in TAC-ABTS values in the CF of sea urchins under HS agrees with findings in the hemolymph of the marine mussel *Mytilus coruscus* [55], in which TAC-ABTS increased after exposure to benzo(α)pyrene and metals. An increase in TAC-ABTS together with ROS and RNS was also found in *P. lividus* following microplastic exposure [42]. The discrepancy observed between the FRAP, which decreases following HS, and the TAC, which, on the contrary, increases following HS, could be due to the different antioxidant components involved in these two analytical methods; in fact, FRAP is predominantly sensitive to low-molecular-weight molecules, while the ABTS method is also affected by phenolic compounds and antioxidants abounding with sulfhydryl groups [56]. However, in *P. lividus* coelomocytes, TAC values did not significantly differ between experimental groups. We do not have a clear explanation for the different behavior of this parameter in CF compared to coelomocytes. In other studies, TAC increased in *P. lividus* coelomocytes following microplastic exposure [42] and in the marine mussel *Mytilus coruscus* hemolymph and tissues after exposure to chemical stressors [55].

Based on the sensitivity of the TAC-ABTS assay to sulfhydryl groups, the rise in TAC-ABTS levels supports the increase in TTL, which, unlike TAC-ABTS, was able to disclose an intense rather than a moderate HS. This biomarker evaluates total thiol (sulfhydryl group, -SH) levels, adding further information on the antioxidant potential of biological samples, since the thiol protein groups, at least in healthy humans, represent approximately 50% of the plasma total antioxidant barrier [32]. In hens exposed for 6 days to pharmacological stress causing immunosuppression, neither TTL nor TAC-ABTS levels were significantly changed at the end of treatment [57]. However, one week later, these biomarkers were found to change in the opposite direction, i.e., TTL decreased whereas TAC-ABTS levels increased [57].

This study represents the first attempt to examine the NO metabolites (NO_x_) in sea urchin CF, demonstrating that they are not influenced by HS. Conversely, the evaluation of the RNS levels in the coelomocytes showed a progressive rise as the temperature increased. This is in agreement with Nash et al. [58] who detected, in the testicular tissue of *Crossostrea virginica* exposed to high temperatures, an increase in nitrotyrosine, an RNS marker. Similarly, in the sea urchin *Arbacia punctulata*, the same research team found a progressive increase in the nitrotyrosine signal as the temperature increased in both female and male gonadal tissues [59]. On the other hand, coelomocytes of *P. lividus* showed no significant changes in nitrite formation detected by the Griess reaction following exposure to acidified seawater [60].

MPO is an enzyme with well-known inflammatory and pro-oxidant activities in mammalian immune cells [61]. In invertebrates, MPO, together with acid and alkaline phosphatases and phenoloxidase, is able to inactivate some pathogens and promote the phagocytosis of immune cells [62,63,64,65]. In the present study, we demonstrated that MPO is affected by HS. This result is in line with the increase in the TAC-ABTS and TTL values, and it may be associated with maintaining the redox balance. A significant increase in MPO activity has also been found in the sea cucumber, *Apostichopus japonicus*, during summer compared to autumn and is associated with aestivation, a torpid state of the animal’s metabolism and, probably, a survival strategy against stress [66]. In invertebrates, MPO’s contribution to immune mechanisms mediated by coelomocytes or biological fluids is unknown. A recently discovered mechanism to combat pathogens by either invertebrate phagocytes or vertebrate leucocytes is the formation of extracellular traps (Ets) [64]. Ets entail the expulsion of chromatin from the nucleus via ROS involvement and can be induced by various agents such as the protein kinase C activator, phorbol myristate acetate, H_2_O_2_, lipopolysaccharide or bacteria [67]. The formation of mammalian Ets depends on enzymes such as neutrophil elastase, MPO, the citrullination of histones and protease activity [68]. This mechanism would indirectly provide a role for MPO in the invertebrate immune response [64].

The protease activity plays a relevant role in the innate immune mechanisms against pathogens [69]. In invertebrates, serine proteases are key effectors of immune response by intervening in the cleaving of the prophenoloxidase zymogen to generate activated phenoloxidase [70]. In the present study, protease activity increased progressively with the increasing temperature. This is in agreement with Fernández-Boo et al. [71] who, in the coelomic fluid of *P. lividus*, detected a significantly higher protease activity in warmer (May-June) than in colder (January) periods of the year.

Different or even opposite trends have been observed in some metabolic functions in sea urchin eggs and coelomocytes. These assessments should be considered in a general context partially influenced by lower viability of the eggs collected from individuals subjected to either moderate (23 °C) or intense (28 °C) HS, a result in agreement with previous research (for review, see [72,73]). In this study, viability was only evaluated in eggs, for which we had a validated protocol, but not in coelomocytes. However, considering that all the evaluations conducted on this cell population were based on cells adhered to the well and, thereby, presumably alive, we do not believe that the viability parameter could have influenced our results.

Mitochondrial activity, evaluated by MMP, provides useful information on the metabolic activity of cells [74]. A reduction in MMP reflects the impairment of cellular function in either immune cells [75,76] or gametes [16,38]. In a previous study [21], by applying a similar experimental design with the sea urchin *P. lividus* collected at the beginning of the breeding season, the MMP of coelomocytes exposed to 23 °C significantly increased after 3 days but was similar to the control after 7 days of HS. In the group exposed to 28 °C, the MMP was lower than the control after either 3 or 7 days of exposure. On the contrary, in the present study, in either coelomocytes or eggs, MMP values in sea urchins exposed to 28 °C were higher than in those exposed at 17 °C. It is difficult to explain this different response that might be attributed to a season-dependent thermotolerance. This occurrence was detected in a study on *M. galloprovincialis* in which animals collected in different seasons (summer, fall and winter) and reproductive stages (quiescence, initiation and full activity) showed different thermotolerance with better resistance to HS in winter compared to the other seasons [77]. However, the different thermotolerance could also be traced back to the different reproductive stages associated with a different metabolism [78].

The production of H_2_O_2_ from mitochondrial or cytosolic compartments is a marker of oxidative stress and immune competence in the gametes [38,40] and in immune cells [79], respectively. In this study, the outcomes in terms of H_2_O_2_ production are similar to those related to mitochondrial activity, in either coelomocytes or eggs, with higher values in the sea urchins exposed to 28 °C than in those exposed to 17 °C. This result differs from previous findings in which, using the same experimental design, a progressive reduction in H_2_O_2_ content in *P. lividus* coelomocytes under HS was detected [21] and suggests the prevalent mitochondrial origin of this H_2_O_2_ production. A different thermotolerance linked to the different seasonal and reproductive periods, as mentioned above, could be the basis of this discrepancy. This result, however, agrees with previous studies showing an increasing production of H_2_O_2_ in the coelomocytes of the earthworm *Eisenia hortensis* exposed for 16 h to increasing temperatures, from 4 to 44 °C [80]. In the eggs, our results in H_2_O_2_ production are in line with findings in the sea urchin *A. punctulata* that, following 7-day exposure to high temperature, showed increasing levels of protein carbonyl, an indicator of oxidative stress, along with increasing temperature [59].

Interestingly, the pH_i_ shows relevant differences between the two cell types. In previous studies, we found a slight decrease in pH_i_ in *M. galloprovincialis* spermatozoa after 7-day exposure to high temperatures [16], and a significant reduction in sperm pH_i_ was found in the ascidian *Ciona robusta* after 1 day of exposure to acidified seawater [47]. Conversely, a significant pH_i_ increase was found in *M. galloprovincialis* sperm after either 3 or 7 days of exposure to acidified seawater [48]. The mechanisms underlying this different pH_i_ behavior in *P. lividus* coelomocytes and eggs exposed to high temperatures are not clear. In the yeast *Saccharomyces cerevisiae*, it was hypothesized that pH_i_ modifications may be related to an increase in thermotolerance [81]. In addition, in two coral species with differing bleaching susceptibilities, *Pocillopora damicornis* (a thermally sensitive coral) and *Montipora capitata* (a thermally resilient coral) exposed to increasing temperature treatments, cells isolated from the former were more prone to decreasing pH_i_ than cells isolated from the latter [82].

In a previous study conducted at the beginning of the breeding season [21], we hypothesized that a 7-day short-term HS could compromise the immune defenses of *P. lividus*. In the present study, conducted three months later in the middle of the breeding season, we found a different situation, in which MPO and protease activity were at their maximum expression together with mitochondrial activity and H_2_O_2_ production. This suggests that, at least in this physiological phase, the sea urchins were fully resilient, contrasting the stressful event with greater thermotolerance.

## 5. Conclusions

The exposure of the sea urchin *P. lividus* to temperatures higher than those optimal for reproductive activity negatively affected both reproductive and immune cells. In the coelomic fluid, the oxidative state was significantly altered, and the enzymatic activities related to the immune defense increased. In coelomocytes and eggs, both the mitochondrial activity and the production of ROS increased; in coelomocytes, RNS increased, while the pH_i_ showed different variations in the two cell types. The analysis of the derivative effects assumes enormous significance in the general framework of the global warming we are experiencing, allowing us to estimate the degree of thermotolerance of a species and to identify proper strategies to take shelter to safeguard the survival of species at risk. The simultaneous analysis of gametes and immune cells provides information on the different behavior of these cells under heat stress. Furthermore, the assessment of some biomarkers of the coelomic fluid may allow us to predict the resilience of these marine organisms under stress to oxidative and immune challenges.

## Figures and Tables

**Figure 1 biomolecules-13-01216-f001:**
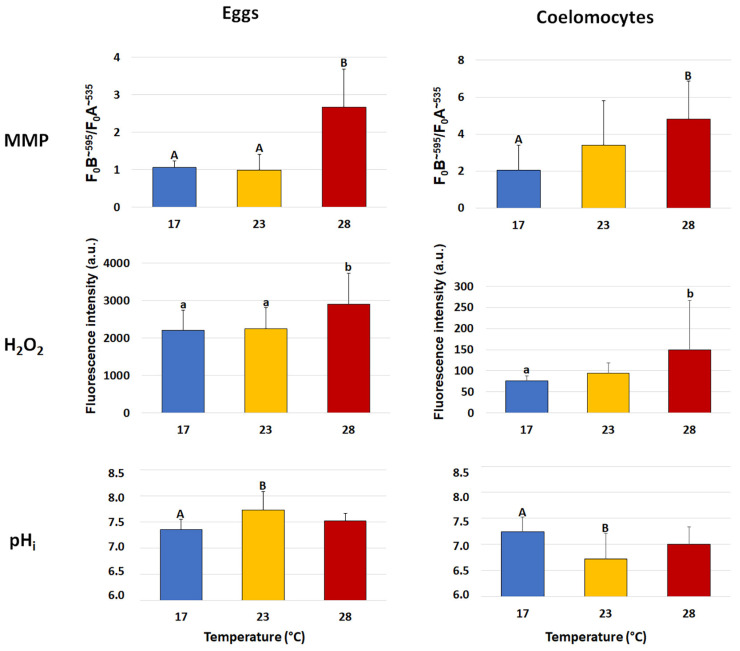
Mean (±SD) levels of mitochondrial membrane potential (MMP), hydrogen peroxide (H_2_O_2_) content and intracellular pH (pH_i_) in the sea urchin *Paracentrotus lividus* eggs and coelomocytes collected from animals exposed to 17 (control), 23 and 28 °C for 7 days. MMP was measured as the ratio between the absolute fluorescence intensity emission peaks at ~595 (F_0_B) and ~535 (F_0_A) nm wavelengths. H_2_O_2_ content was evaluated in arbitrary units (a.u.) on fluorescence emission peak intensity at ~525 nm wavelength. Capital and small letters indicate values that differ significantly at *p* < 0.01 and *p* < 0.05, respectively, according to LSD’s pairwise comparison test.

**Figure 2 biomolecules-13-01216-f002:**
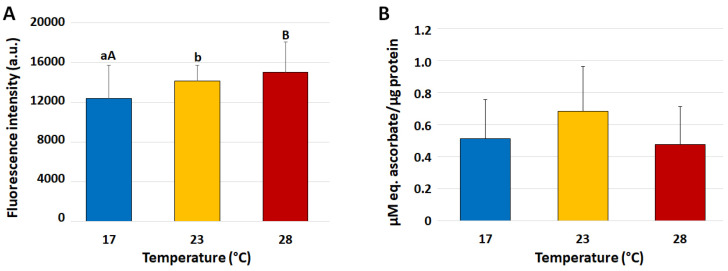
Mean (±SD) levels of intracellular reactive nitrogen species (RNS) (**A**) and total antioxidant capacity (TAC ABTS method) (**B**) in *Paracentrotus lividus* coelomocytes collected from animals exposed to 17 (control), 23 and 28 °C for 7 days. Capital and small letters indicate values that differ significantly at *p* < 0.01 and *p* < 0.05, respectively, according to LSD’s pairwise comparison test.

**Table 1 biomolecules-13-01216-t001:** Celomatic fluid (CF) pH (pH_CF_) and osmolarity, coelomocyte concentration and total eggs collected in the sea urchins exposed for 7 days to 17, 23 and 28 °C.

		17 °C	23 °C	28 °C
pH_CF_		7.95 ± 0.05	7.88 ± 0.10	7.90 ± 0.06
CF Osmolarity	mOsm	1121 ± 3	1122 ± 5	1121 ± 5
Coelomocyte concentration	×10^6^ mL^−1^	14.8 ± 7.9 **a**	24.0 ± 7.9 **b**	14.0 ± 7.9 **a**
Total eggs	×10^3^	387 ± 143	337 ± 188	370 ± 143

Different letters indicate values that differ significantly at *p* < 0.05 according to LSD’s pairwise comparison test.

**Table 2 biomolecules-13-01216-t002:** Mean (± SD) values of coelomic fluid biomarkers related to oxidative profile and protease activity in *Paracentrotus lividus* individuals (n. 18) exposed to 17, 23 and 28 °C for 7 days.

	17 °C	23 °C	28 °C
FRAP (μM)	74.2 ± 15.1 **a**	66.1 ± 12.1	58.8 ± 8.2 **b**
TAC-ABTS (μM)	93.5 ± 61.2 **a**	188.2 ± 125.0 **b**	148.8 ± 46.7
TTL (µM)	48.2 ± 14.9 **A**	52.3 ± 25.8 **A**	104.9 ± 25.7 **B**
NO_x_ (μM)	45.3 ± 16.3	43.0 ± 5.0	51.3 ± 20.5
MPO (OD_450 nm_)	8.5 ± 3.9 **a**	14.0 ± 5.3	15.2 ± 6.0 **b**
Protease activity (%)	4.5 ± 2.9 **Aa**	10.4 ± 6.7 **bC**	21.0 ± 4.5 **BD**

FRAP: total antioxidant capacity (FeSO_4_•7H_2_O equivalents); TAC: ABTS-based total antioxidant capacity (ascorbic acid equivalents); TTL: total thiol levels; NO_x_: nitric oxide metabolites (NaNO_3_ equivalents); MPO: myeloperoxidase (optical density, OD); Protease activity (% trypsin activity). Capital (A vs. B and C vs. D) and small (a vs. b) letters indicate values that differ significantly at *p* < 0.01 and *p* < 0.05, respectively, according to LSD’s pairwise comparison test.

## Data Availability

Not applicable.

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
