# Peer review of "Immune and Reproductive Biomarkers in Female Sea Urchins Paracentrotus lividus under Heat Stress"

_biomolecules, 2023, doi:10.3390/biom13081216_

Round 1

Reviewer 1 Report

The subject is very interesting and increases knowledge in the scientific field. The paper contains a considerable amount of data, which will contribute to establishing mitigation strategies for marine species. Nevertheless, the results section is a bit confusing and conclusions seem the continuity of the discussion. Some minor English grammar and spelling inconsistencies are found throughout the text, so I advise the authors to carefully revise the English.

Please see my specific comments:

Title - I think the title must be changed as throughout the paper immune biomarkers are given more importance than reproduction. Maybe: Immune and reproductive biomarkers ... with this title I was expecting more emphasis on reproduction.

Lines 76-78 - if females are better protected, why not use males
M&M - I think it is relevant to state the maturity
stage of the individuals.
Line 127 - The ferric reducing ability of plasma (FRAP) assay - ferric reducing power assay?
Table 1 is misplaced
Lines 254-258 - Are the statistical analysis also performed between pHCF from the different temperatures and seawater pH ?
Please clarify.
 I would prefer to see the statistical data with the test values and the degrees of freedom. I do not think p values alone bring relevant information. Please add the required info in lines: 256, 260,268,270,274,275, 277, 280, 292, 295, 301, 302, 308, 311, 316 and 317.
Lines 269-278 - please remove from the description of the results the values already reported in Table 2. The text gets hard to read, and there is no need to double this info.
Lines 270-271: not clear whether that is versus control
Line 277- "further growed"- odd sentence. Please change it.
3.3 section - The redaction is a bit confusing. 17ºC is described as control or as 17ºC
Table 2 lower and uppercase letters describe the p-value already mentioned in the text, and, in my opinion, it is not a piece of relevant information in the table caption. Personally, I would like to see letters depicting the differences. For example, different lowercase letters represent significant statistical differences, or "a" shows differences with control, b differences between…
Lines 292-297 and 327-328 have the same comment regarding the lower and upper case letters and p values.
Lines 299-300- I do not think it is a similar trend. We cannot see this by looking at the graphs and the letters in each column.
The similarity is only between the control and 28ºC. Please rephrase.
Figure 1 - The same comment regarding the letters in the columns,  figure caption and graphs.
Lines 315-320 - Are all treatments different from each other?
See the comments regarding the letters and caption.
Lines 327-328 - Please see the previous comments regarding the lower and upper case letters and p values.
Discussion:
The discussion is too long, and the authors should remove all descriptions of the results and focus on explaining their findings.
Lines 344-345- I think this sentence deserves an explanation about the similarity of the CF and seawater
Line 347 - change to: have not previously been performed
This sentence is confusing. Please rephrase and clarify it. Contrary to the studies with eggs there are studies with sperm, however both eggs and sperm count seemed not to be affected.
Lines 343- 357 - I found this paragraph confusing. Please improve it.
line  389 - change “, that,”  by “, which,”
line 393- are you saying the TAC was not affected in this study?
It is not true since increased at 23ºC.
In lines 379-381, you stated that FRAP differences in your studies might be due to the exposure duration, as in a previous study, FRAP increase was recorded after several hours and depletion after 30 days.
Line 399 - in this sentence: “This biomarker evaluates of total thiol…” please remove "of"
Line 417, 423 - 425-  And the MPO function in invertebrates? Please see this paper or other more recent papers.
“It has been reported that in invertebrates, ACP, ALP, PO and MPO are able to inactivate some pathogens and promote the phagocytosis of immune cells (Holmblad & Soderhall 1999; Cerenius & Soderhall 2004; Xing, Lin & Zhan 2008) in  https://doi.org/10.1111/are.13005
Lines 431-432 - and protease activity?
Line 449- I would like to see an explanation of the physiological relevance/consequence of the alteration of the MMP.
Lines 455-456 and 469-470- Again, it is crucial that the authors clarify the maturity stage or the season of this assay.
Lines 462-464 and 470—471 are it possible to track back, but which is the probable stage of maturity of the sea urchins in spring?
From previous studies of the team, it is possible to know the probable maturity stage, and I think it is relevant to say something about it. I already commented on this on lines 96-97
I missed the whole picture approach in the discussion.
As previously mentioned, I think the authors must avoid repeating the description of the results and directly discussing it.
I would like to see the role of mitochondria in the immune system
and how ROS are related to this function, specifically H2O2.

Conclusions
The conclusions must focus only on the results of the current study. Any comparison with previous studies must be moved to the discussion. Please pay special attention to lines 504-506
505-506- This belongs to the discussion and must be moved to the discussion section. Throughout the discussion, the differences between studies were related to different seasons or breeding stages but were never specified. This must be explained and explored in the discussion section, not here.
Line 503- The following information is unnecessary: "which enables to evaluate its redox state, inflammation, and non-specific immunity,... " Please remove it. The same is true for lines 504-506
Conclusions must be brief and contain the most relevant findings.

Some minor English grammar and spelling inconsistencies are found throughout the text, so I advise the authors to carefully revise the English.

Author Response

The subject is very interesting and increases knowledge in the scientific field. The paper contains a considerable amount of data, which will contribute to establishing mitigation strategies for marine species. Nevertheless, the results section is a bit confusing and conclusions seem the continuity of the discussion. Some minor English grammar and spelling inconsistencies are found throughout the text, so I advise the authors to carefully revise the English.

Reply. We really appreciated the analysis conducted on our manuscript and thank the reviewer for her/his kind efforts. We trust that by resolving these criticisms we will be able to improve the quality of our paper and clarify the doubts raised.

Please see my specific comments:

Title - I think the title must be changed as throughout the paper immune biomarkers are given more importance than reproduction. Maybe: Immune and reproductive biomarkers ... with this title I was expecting more emphasis on reproduction.

Reply. We agree with this suggestion and changed the title accordingly.

Lines 76-78 - if females are better protected, why not use males

Reply. Actually, one of the reasons that led us to choose females is precisely linked to the higher immune reactivity of this gender, allowing for better detection of any variations. Furthermore, isolating a gender, on the basis of the information present in the literature, would have allowed us to reduce the amount of individual variability and obtain greater statistical power. Then, we had already done a study on the same topic, with different evaluations, using P.lividus females (Murano et al., Animals 2023); this second study would have resulted in confirmation and deepening of what we have already found. Finally, we had already carried out a study on the effect of thermal stress in the male Mytilus galloprovincialis; we also wanted to evaluate HS response based on some functional parameters in the female gametes.

M&M - I think it is relevant to state the maturity stage of the individuals.

Reply. We evaluated this parameter and reported this information in the text (L97-99)

Line 127 - The ferric reducing ability of plasma (FRAP) assay - ferric reducing power assay?

Reply. Yes, it's the same method. Naturally, in our case, we used a method developed on blood plasma using another organic matrix such as the coelomic fluid. Similar adaptations are reported in the literature on other biological matrices of both animal and vegetable origins. However, considering that the indication of the term "plasma" in the acronym could generate confusion, we report the method as you suggested.

Table 1 is misplaced
Reply. Done

Lines 254-258 - Are the statistical analysis also performed between pHCF from the different temperatures and seawater pH ? Please clarify.
Reply. The statistical analysis was also performed between pHCF from the different temperatures and seawater pH, we clarify this in the text (L106-107).

I would prefer to see the statistical data with the test values and the degrees of freedom. I do not think p values alone bring relevant information. Please add the required info in lines: 256, 260,268,270,274,275, 277, 280, 292, 295, 301, 302, 308, 311, 316 and 317.
Reply. Sorry, we apologize but we did not quite understand your suggestion. In all the cases that you referred p values were always supported by test values. In the case of L256, for example, test values were 8.19 ± 0.06 (seawater pH) vs. 7.95 ± 0.05, 7.88 ± 0.10, and 7.90 ± 0.06 (pH of the three groups, as reported in the Table 1). Regarding L260, test values were reported in both the Table 1 and the text (24.0 ± 7.9 vs. 14.8 ± 7.9 and 14.0 ± 7.9 x 106 mL-1). And so, on. If the reviewer refers to the exact value of p, we think that, reporting in the text a value of p < 0.023 instead of p < 0.05, we would create inconsistencies between the text and the tables where, to standardize the data, we considered a significance level of p at 0.05 and 0.01.

The number of cases -1 gives the degrees of freedom (DF). Our study was conducted on 3 groups of 6 individuals each, therefore, the total variance presents 18-1= 17 DF, while the variance between groups (3-1) = 2 and the variance within groups (or error) (6-1) x 3 = 15 DF. For eggs, these data were reduced by one in all groups because one individual in each group did not release gametes. Also, although the data were evaluated in duplicate or triplicate, we used to analyze the averages of the duplicate or triplicate results. We added this information in Statistical analysis section on L245.

Lines 269-278 - please remove from the description of the results the values already reported in Table 2. The text gets hard to read, and there is no need to double this info.

Reply. Done

Lines 270-271: not clear whether that is versus control

Reply. Done

Line 277- "further growed"- odd sentence. Please change it.

Reply. We modified the text accordingly

3.3 section - The redaction is a bit confusing. 17ºC is described as control or as 17ºC

Reply. We modified the text accordingly

Table 2 lower and uppercase letters describe the p-value already mentioned in the text, and, in my opinion, it is not a piece of relevant information in the table caption. Personally, I would like to see letters depicting the differences. For example, different lowercase letters represent significant statistical differences, or "a" shows differences with control, b differences between…

Reply. Sorry, in the first version of the MS, we used the uppercase font to fit the table within its space, which, then, we forgot to remove this format. We modified the text accordingly.

Lines 292-297 and 327-328 have the same comment regarding the lower and upper case letters and p values.
Reply. We modified the text accordingly

Lines 299-300- I do not think it is a similar trend. We cannot see this by looking at the graphs and the letters in each column. The similarity is only between the control and 28ºC. Please rephrase.

Reply. We agreed and modified the text accordingly

Figure 1 - The same comment regarding the letters in the columns,  figure caption and graphs.
Reply. We agreed and modified the text accordingly

Lines 315-320 - Are all treatments different from each other? See the comments regarding the letters and caption.
Reply. Regarding RNS values in coelomocytes, as shown in Figure 2A, the group at 17 °C significantly differed from the group at 23 °C (p < 0.05). Moreover, the group at 17 °C significantly differed from the group at 28 °C (p < 0.01). However, there were no significant differences between the group at 23 and 28 °C. In the case of TAC in coelomocytes (Fig. 2B), there were no significant differences between groups.

Lines 327-328 - Please see the previous comments regarding the lower and upper case letters and p values.
Reply. Done

Discussion:
The discussion is too long, and the authors should remove all descriptions of the results and focus on explaining their findings.
Reply. We removed the description of the results. However, due to the many clarifications requested and the number of results to be discussed, we were unable to reduce the size of the Discussion.

Lines 344-345- I think this sentence deserves an explanation about the similarity of the CF and seawater

Reply. We clarify this aspect.

Line 347 - change to: have not previously been performed

Reply. Done

This sentence is confusing. Please rephrase and clarify it. Contrary to the studies with eggs there are studies with sperm, however both eggs and sperm count seemed not to be affected.

Reply. We agreed and modified the text accordingly

Lines 343- 357 - I found this paragraph confusing. Please improve it.

Reply. We agreed and modified the text accordingly

line  389 - change “, that,”  by “, which,”

Reply. Done

line 393- are you saying the TAC was not affected in this study? It is not true since increased at 23ºC.

Reply. Yes, in coelomocytes, as shown in Figure 2B, we did not find significant differences between groups in relation to TAC values.

In lines 379-381, you stated that FRAP differences in your studies might be due to the exposure duration, as in a previous study, FRAP increase was recorded after several hours and depletion after 30 days.
Reply. We confirm it and, for a better clarity, we modified the text accordingly

Line 399 - in this sentence: “This biomarker evaluates of total thiol…” please remove "of"
Reply. Done

Line 417, 423 - 425-  And the MPO function in invertebrates? Please see this paper or other more recent papers.
“It has been reported that in invertebrates, ACP, ALP, PO and MPO are able to inactivate some pathogens and promote the phagocytosis of immune cells (Holmblad & Soderhall 1999; Cerenius & Soderhall 2004; Xing, Lin & Zhan 2008) in  https://doi.org/10.1111/are.13005
Reply. Done

Lines 431-432 - and protease activity?
Reply. Done

Line 449- I would like to see an explanation of the physiological relevance/consequence of the alteration of the MMP.
Reply. Done

Lines 455-456 and 469-470- Again, it is crucial that the authors clarify the maturity stage or the season of this assay.

Reply. We added this information (L97-99). As reported in L88, we collected these animals in the middle of the breeding season (April-May) for this species while in the previous study (Murano et al., Animals 2023) the animals were analyzed at the beginning of the breeding season (January-February). In this first study, we tried to recover the eggs but without success, although in previous years this was already possible in the same period. Unfortunately, due to climate change, the reproductive cycle of these animals is changing.

Lines 462-464 and 470—471 are it possible to track back, but which is the probable stage of maturity of the sea urchins in spring?

Reply. See above

From previous studies of the team, it is possible to know the probable maturity stage, and I think it is relevant to say something about it. I already commented on this on lines 96-97

Reply. See above

I missed the whole picture approach in the discussion.

As previously mentioned, I think the authors must avoid repeating the description of the results and directly discussing it.
I would like to see the role of mitochondria in the immune system
and how ROS are related to this function, specifically H2O2.

Reply. See above

Conclusions
The conclusions must focus only on the results of the current study. Any comparison with previous studies must be moved to the discussion. Please pay special attention to lines 504-506
505-506- This belongs to the discussion and must be moved to the discussion section.

Reply. We agreed and modified the text accordingly

Throughout the discussion, the differences between studies were related to different seasons or breeding stages but were never specified. This must be explained and explored in the discussion section, not here.

Reply. We agreed and modified the text accordingly

Line 503- The following information is unnecessary: "which enables to evaluate its redox state, inflammation, and non-specific immunity,... " Please remove it. The same is true for lines 504-506
Reply. Done

Conclusions must be brief and contain the most relevant findings.

Reply. We agreed and modified the text accordingly

P.S. Regarding starvation during the week of the experiment, talking with the animal facility staff we discovered (and this was confirmed in the animal facility activity log) that, due to a mistake, a meal had been fed to the animals halfway through the experimental period. We have inserted the correct information in the text (L107-114). We are sorry for the mistake.

Reviewer 2 Report

This paper provided some interesting data about stress responses in immune and reproductive systems of sea urchins to acute heat stress and pointed out several useful biomarkers to show functional alterations in cells caused by the stress. The graphs are clear and the writing is well prepared. However, I still have the following questions that needs the authors to give some discussion or improvement in the manuscript:

 1 In the experimental design, during the experimental period for 7 days, the animals were not fed, this starving may become another stress especially for animals under high temperature. It is better to discuss a little bit about possible effects of starving stress on examined parameters.

 2 The authors did not provide body weight data as a indicator of developmental and physiological condition of the animals. I suggest supplement of body weight data of the experimental animals.

Author Response

This paper provided some interesting data about stress responses in immune and reproductive systems of sea urchins to acute heat stress and pointed out several useful biomarkers to show functional alterations in cells caused by the stress. The graphs are clear and the writing is well prepared. However, I still have the following questions that needs the authors to give some discussion or improvement in the manuscript:

Reply. We thank the reviewer for the appreciation of our study and for the criticisms raised. We trust that by resolving these criticisms we will be able to improve the quality of our paper and clarify the doubts raised.

1 In the experimental design, during the experimental period for 7 days, the animals were not fed, this starving may become another stress especially for animals under high temperature. It is better to discuss a little bit about possible effects of starving stress on examined parameters.

Reply. We agree with the reviewer and considered this aspect in the experimental project. However, due to a mistake, we have partially avoided this second source of stress. In fact, in the experimental project, we had agreed with the animal facility staff that the animals would not receive the food (Ulva spp.) in the week of experimental treatment (from Friday to Friday). However, due to a misunderstanding, which emerged when we met and discussed the reviews, the staff referred (and this was confirmed in the activity diary of our facility) that they had fed the animals on day 3 of treatment (Monday). We are very sorry to have to report this error which, however, unintentionally weakened the source of stress related to the starvation. We commented on this information on L107-114.

2 The authors did not provide body weight data as a indicator of developmental and physiological condition of the animals. I suggest supplement of body weight data of the experimental animals.

Reply. Sorry, we had this data but we missed adding this information. We added both the test diameters and the total weights of the animals enrolled in this study on L115-117. We also add information on the assessment of their sexual maturity.